# A semi-automated machine-learning based workflow for ellipsoid zone analysis in eyes with macular edema: SCORE2 pilot study

**Tyler Etheridge[1], Ellen T. A. Dobson[2], Marcel Wiedenmann[3], Chandana Papudesu[1], Ingrid U. Scott[4], Michael S. Ip[5], Kevin W. Eliceiri[2], Barbara A. Blodi[1], Amitha Domalpally[1] ***

**1** Department of Ophthalmology and Visual Sciences, University of Wisconsin School of Medicine and Public Health, Madison, WI, United States of America, **2** Laboratory for Optical and Computational Instrumentation, University of Wisconsin-Madison, Madison, WI, United States of America, **3** KNIME GmbH, Konstanz, Germany, **4** Departments of Ophthalmology and Public Health Sciences, Penn State College of Medicine, Hershey, PA, United States of America, **5** Doheny Eye Institute, University of California Los Angeles Stein Eye Institute, Los Angeles, CA, United States of America

* domalpally@wisc.edu

**Data Availability Statement:** The KNIME workflow is available at https://hub.knime.com/tetheridge/

## Abstract

### Background and objective

To develop a semi-automated, machine-learning based workflow to evaluate the ellipsoid zone (EZ) assessed by spectral domain optical coherence tomography (SD-OCT) in eyes with macular edema secondary to central retinal or hemi-retinal vein occlusion in SCORE2 treated with anti-vascular endothelial growth factor agents.

### Methods

SD-OCT macular volume scans of a randomly selected subset of 75 SCORE2 study eyes were converted to the Digital Imaging and Communications in Medicine (DICOM) format, and the EZ layer was segmented using nonproprietary software. Segmented layer coordinates were exported and used to generate *en face* EZ thickness maps. Within the central subfield, the area of EZ defect was measured using manual and semi-automated approaches via a customized workflow in the open-source data analytics platform, Konstanz Information Miner (KNIME).

### Results

A total of 184 volume scans from 74 study eyes were analyzed. The mean±SD area of EZ defect was similar between manual (0.19±0.22 mm$^2$) and semi-automated measurements (0.19±0.21 mm$^2$, p = 0.93; intra-class correlation coefficient = 0.90; average bias = 0.01, 95% confidence interval of limits of agreement -0.18–0.20).

### Conclusions

A customized workflow generated via an open-source data analytics platform that applied machine-learning methods demonstrated reliable measurements of EZ area defect from *en*

spaces/Public/latest/Raw_Thickness_Maps. The KNIME workflow protocol and training dataset are available at https://doi.org/10.6084/m9.figshare.11774577.

**Funding:** "The Standard Care vs Corticosteroid for Retinal Vein Occlusion (SCORE) 2 Study was supported by National Eye Institute (National Institutes of Health, Department of Health and Human Services) grants U10EY023529, U10EY023533, and U10EY023521; and Allergan, Inc. The funder (Allergan, Inc.) had no role in study design, data collection and analysis, decision to publish, or preparation of the manuscript. KNIME GmbH provided support for this study in the form of salary for MW. The specific roles of all authors are articulated in the 'author contributions' section. KNIME GmbH had no role in study design, data collection and analysis, decision to publish, or preparation of the manuscript.

**Competing interests:** The authors have read the journal's policy and the authors of this manuscript have the following competing interests: MW is a paid employee of KNIME GmbH. Additionally, the funder Allergan, Inc. provided support for this study. This does not alter our adherence to PLOS ONE policies on sharing data and materials. There are no patents, products in development, or marketed products to declare.

*face* thickness maps. The result of our semi-automated approach were comparable to manual measurements.

## Introduction

Optical coherence tomography (OCT) scans are routinely used in clinical practice for monitoring therapeutic efficacy in patients with macular edema. Central retinal thickness measurements obtained from OCT are a key outcome measure for clinical trials; however, central retinal thickness is not considered a biomarker or a predictor for visual recovery because of its weak correlation with visual acuity (VA).[1] The ellipsoid zone (EZ), previously named the photoreceptor inner segment-outer segment junction, is visualized as a hyperreflective layer in the outer retina on spectral domain optical coherence tomography (SD-OCT) images.[2] SD-OCT enables *in vivo* high-resolution, cross-sectional imaging of the retina, permitting qualitative and quantitative assessment of EZ integrity, which has been correlated with VA in multiple retinal diseases,[3, 4] including retinal vein occlusion (RVO).[5–7]

Many studies examining the EZ layer are qualitative, describing abnormalities of the layer based on signal intensity.[8, 9] Current methods for quantitative assessment of the EZ involve segmentation of the retinal layers using manual or semi-automated techniques.[10, 11] This generates EZ layer thickness measurements with an Early Treatment Diabetic Retinopathy Study (ETDRS) grid. An alternative and more intuitive method would be to assess the area of EZ defect or loss using *en face* thickness maps created from the segmented layers.[5] The *en face* maps are two dimensional and present areas of EZ absence, i.e. very low thickness areas are dark regions against a brighter background of normal EZ thickness. Manual segmentation of the EZ is laborious and time intensive.[12] Therefore, we sought to utilize accessible, machine-learning methods to facilitate the identification of these regions. We trained a classifier model using the Trainable Weka Segmentation plugin[13] of Fiji [14, 15], an open-source image processing software package specifically designed for scientific image analysis, to detect the areas of EZ defect. Using this trained classifier, we then applied it within a novel, customized workflow using the open-source software platform Konstanz Information Miner (KNIME)[16] to evaluate the EZ assessed by SD-OCT generated *en face* thickness maps in eyes with macular edema secondary to central retinal vein occlusion (CRVO) or hemi-retinal vein occlusion (HRVO).

## Methods

### Participants

Study data were obtained from the **S**tudy of **CO**mparative Treatments for **RE**tinal Vein Occlusion **2** (SCORE2), a multicenter, prospective, randomized non-inferiority trial of eyes with macular edema secondary to CRVO or HRVO comparing intravitreal anti-vascular endothelial growth factor agents (anti-VEGF) bevacizumab vs. aflibercept (Clinicaltrials.gov identifier NCT01969708).[17] The study was approved by institutional review boards (IRB) associated with each center (University of Wisconsin Madison IRB Number 2014-0256-CR006) and adhered to the tenets of the Declaration of Helsinki. All participants provided written informed consent. The SCORE2 design and methods have been previously described in detail. [18] In summary, 362 participants were randomized to receive either intravitreal bevacizumab or aflibercept. The study visits were conducted per protocol with treatment provided per protocol from baseline through month 12, and then at the discretion of the investigator thereafter.

Inclusion criteria were center-involved macular edema defined as central subfield thickness of ≥300 μm (or ≥320 μm if measured with Heidelberg Spectralis Machine).

Seventy-five participants were randomly selected from the SCORE2 baseline dataset, which represented approximately 20% of the total number of trial subjects. Participants were stratified based on baseline VA into three groups, including good (73–59 letters: 20/40-20/63), moderate (58–49 letters: 20/80-20/100), and poor (48–19 letters: 20/125-20/400). Twenty-five subjects were randomly selected from each stratum.

## SD-OCT image acquisition

All SD-OCT images were acquired by certified technicians using the SCORE2 reading center (Fundus Photograph Reading Center, University of Wisconsin) approved protocol with either Carl Zeiss Meditec Cirrus (Carl Zeiss Meditec, Dublin, CA) or Heidelberg Spectralis (Spectralis Heidelberg Engineering, Heidelberg, Germany) OCT machine.[18] The Zeiss macular volume scans were 6 mm and comprised of 512 A-scans and 128 B-scans, and the Heidelberg scans were 20 x 20 degrees and comprised of 512 A-scans and 97 B-scans. SD-OCT images were evaluated at baseline, month 1, month 6, and month 12 for all participants.

## Segmentation of EZ layer

The SD-OCT macular volume scans were received in proprietary formats at the central reading center and converted to Digital Imaging and Communications in Medicine (DICOM) format.[19] The EZ layer was segmented in the central subfield (CSF) using custom segmentation software developed using MATLAB (The Mathworks Inc, Natick, Massachusetts, USA.).[20] The CSF consisted of 17 (Spectralis) or 23 (Cirrus) B-scans. The EZ layer is typically visible as a hyperreflective line between the external limiting membrane and the retinal pigment epithelium (RPE). The inner border of the second outer hyperreflective band (EZ layer) and the inner border of the third outer hyperreflective band (RPE) were selected as the EZ layer boundaries (**Fig 1A**).[2]

## Generation of *En face* thickness maps

EZ layer xy coordinates were exported as Extensible Markup Language (XML) format, and those files were used to generate *en face* thickness maps for selected layers via linear interpolation within a customized workflow in the open-source data analytics platform KNIME, version 3.7.2 (**Fig 1B**). Areas of EZ defect appear as dark areas on the thickness maps compared to bright areas with normal EZ.

## Ellipsoid zone area analysis

A machine-learning tool, the Trainable Weka Segmentation (TWS) plugin, was used to generate a classifier to segment regions of EZ defect within generated thickness maps automatically (**Fig 1C and 1D**). This tool is a Fiji plugin that combines a collection of machine-learning algorithms with a set of selected image features to produce pixel-based segmentations.[13] Default segmentation settings in TWS were applied, though the maximum sigma was set to 32. The training features included were Gaussian blur, Hessian, Membrane projections (directional filtering), Sobel filter, Difference of gaussians, and Variance. The classifier applied was Fast Random Forest, a multi-threaded version of random forest by Fran Supek.[21] An estimated subset (~10%) representing the heterogeneity of the thickness maps of the entire dataset were selected to train the classifier to detect areas of EZ defect. Once trained, where the appropriate regions were reliably and reproducibly segmented, the classifier was then applied to the larger

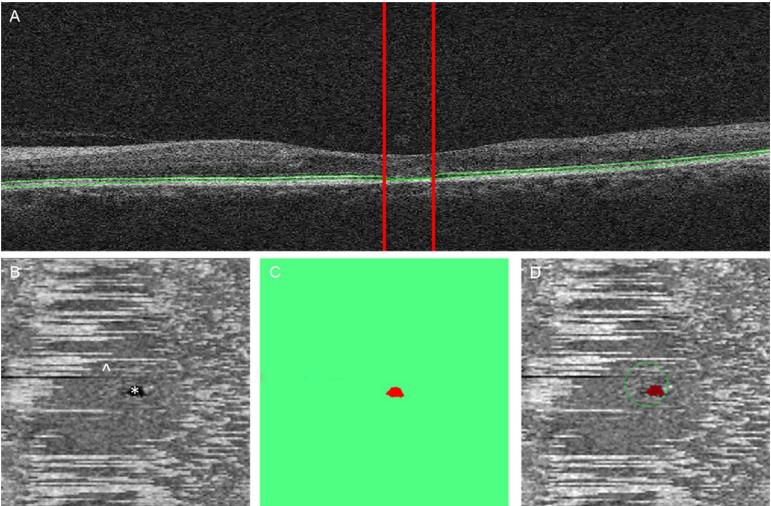

**Fig 1. Semi-automated analysis of ellipsoid zone (EZ) defect using *en face* approach. A.** B-scan with segmentation lines (green) from the top of the EZ layer to the top of the retinal pigment epithelium highlighting EZ defect (vertical red lines). **B.** Distances between segmented lines were linearly interpolated to form *en face* thickness map showing intact (white/grey pixels ^) and defective (black/dark pixels *) EZ. **C.** Application of Trainable Weka Segmentation (TWS) classifier to entire thickness map identifies regions of intact (green) and absent (red) EZ. **D.** The area of absent EZ (red) within the central subfield (green circle) was measured on the *en face* thickness map via the KNIME workflow.

dataset for automatic segmentation and area measurements via the customized KNIME workflow (**Fig 2**). All scans were obtained from the SCORE2 dataset and were therefore from eyes affected by either CRVO or HRVO. The KNIME workflow was applied using a standard

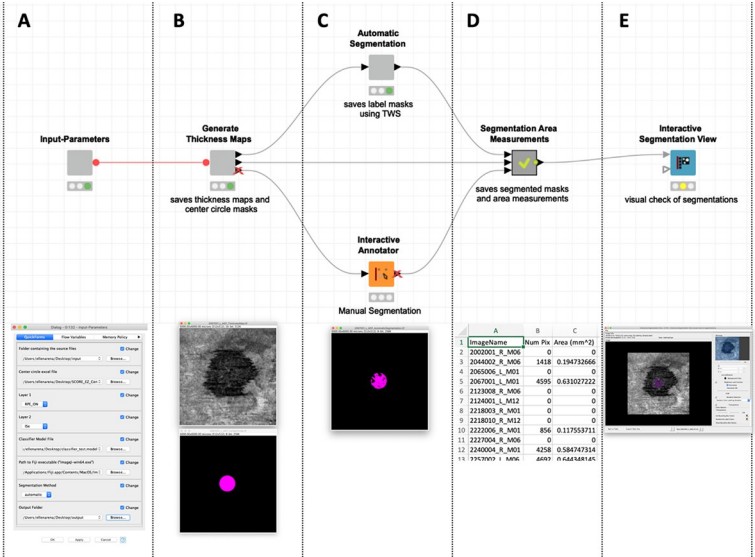

**Fig 2. KNIME workflow.** There are 5 distinct sections (metanodes) of the customized KNIME workflow including: (**A**) selection of input parameters, which includes a source folder containing .xml layer files, an .xls file including xy coordinates for central subfield (CSF) positions, the selection of two layers for thickness map generation, a classifier . model file generated via the Trainable Weka Segmentation (TWS) plugin of Fiji, the path to local Fiji installation, the selection of the segmentation method (either 'automatic' or 'manual'), and the output folder for all results files; (**B**) generation of the *en face* thickness maps and masks of CSF regions; (**C**) either 'automatic', using the TWS classifier, or 'manual' segmentations; (**D**) segmentation area measurements, which are calculated only within the CSF region; and finally (**E**) an interactive segmentation view for a visual check of all segmentations made via the workflow.

desktop computer (Processor: Intel® Core™ i5-45900 CPU @ 3.30 GHz; Memory: 8.00 GB; System Type: 64-bit Operating System).

For manual measurements, the EZ boundary was manually traced within the KNIME workflow, and only the areas of EZ defect within the CSF were quantified. CSF xy coordinates were exported from the custom MATLAB software and used to overlay the CSF regions unique to each image (**Fig 2B**). All thickness maps and area measurements were generated via the same methods for both automatic and manual approaches. The only difference between the semi-automated and manual workflows was the segmentation method applied (**Fig 2C**). The KNIME workflow allowed visual inspection of all segmented regions with image overlays of the area of EZ defect on the *en face* thickness map (**Fig 2E**). All images were analyzed and reviewed by two masked graders and study authors (T.E. and C.P.). Graders reviewed images independently and were masked to segmentation results. Inter-rater reliability and agreement were assessed for manual EZ defect measurements.

The minimum area of EZ defect was defined as 0.004 mm$^2$ and the maximum area of EZ defect was 0.78 mm$^2$ (based on the area of the CSF). The minimum area was selected based on the lowest limit of area measurability used within reading center grading protocols (e.g. drusen circle $C_0$ established by the Age-Related Eye Disease Study (AREDS) Research Group).[22]

## Statistical analysis

We investigated the reliability and agreement of the manual and semi-automated approaches for determining the area of EZ defect within the CSF. Reliability was determined by calculating the intra-class correlation coefficient (ICC).[23] Agreement was determined by calculating the average bias between the manual and semi-automated measurements using the Bland-Altman method.[24] If the average bias does not exceed the 95% confidence interval (CI) of the limits of agreement (LOA), then the methods do not disagree and can be used interchangeably.[25] ICC and average bias were calculated to assess inter-rater reliability and agreement for manual EZ defect measurements. Area measurements were not normally distributed. Therefore, the non-parametric Wilcoxon Signed-Ranks Test was used to compare the area measurements between semi-automated and manual methods. A two-tailed p-value less than 0.05 was considered significant for all hypothesis testing. Statistical analysis was performed using R v3.6.1 (R Foundation for Statistical Computing, Vienna, Austria).

## Results

Of the 75 randomly selected study eyes, one did not have follow-up images after baseline, resulting in 74 study eyes available for analysis. Assessment of the EZ layer was performed at baseline, month 1, month 6, and month 12; however, EZ assessment was not possible on baseline scans due to a >90% rate of ungradable images resulting from signal blockage by hemorrhage or fluid (**Fig 3**). Therefore, only SD-OCT images at months 1, 6, and 12 were analyzed. Of the selected study visits, SD-OCT images were missing from 7 study visits and the retinal layers were not visible for segmentation in 31 eyes due to signal blockage from hemorrhage or fluid (21 at month 1, 6 at month 6, and 4 at month 12), resulting in a total of 184 gradable volume scans for analysis (53 at month 1, 65 at month 6, and 66 at month 12).

Using semi-automated measurements, EZ defect was seen in 36 of 53 thickness maps (67.9%) at month 1, 27 of 65 (41.5%) at month 6, and 29 of 66 (43.9%) at month 12. Combining all time points for a total of 184 images, 92 (50.0%) had an EZ defect area. The mean±SD area of EZ defect as measured by the semi-automated approach was 0.23±0.25 mm$^2$ at month 1 (range 0.005–0.76 mm$^2$), 0.21±0.21 mm$^2$ at month 6 (range 0.007–0.76 mm$^2$), and 0.10±0.15

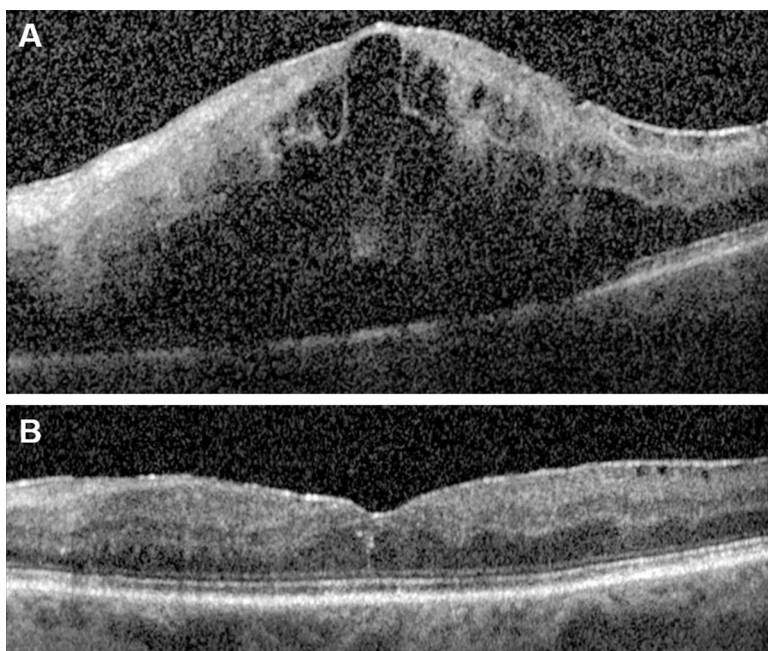

**Fig 3.** (**A**) Representative center point SD-OCT scan deemed ungradable at baseline for ellipsoid zone (EZ) assessment. EZ was considered ungradable when the retinal pigment epithelium was not visible due to signal blockage by hemorrhage and edema. (**B**) Month 1 follow-up SD-OCT scan after single intravitreal anti-VEGF injection. EZ considered gradable.

mm$^2$ at month 12 (range 0.005–0.66 mm$^2$) (**Table 1**). The mean±SD area of EZ defect for all time points combined was 0.19±0.21 mm$^2$ (range 0.005–0.76 mm$^2$), respectively.

The manual approach revealed a mean±SD EZ area defect measurement of 0.24±0.24 mm$^2$ at month 1 (range 0.005–0.78 mm$^2$), 0.19±0.22 mm$^2$ at month 6 (range 0.006–0.78 mm$^2$), and 0.12±0.18 mm$^2$ at month 12 (range 0.005–0.78 mm$^2$). The mean±SD area measurement for all time points combined was 0.19±0.23 mm$^2$ (range 0.005–0.78 mm$^2$). The area of EZ defect combining all time points was indistinguishable between the semi-automated and manual measurements (p = 0.93) (**Fig 4A**).

The ICC between semi-automated and manual measurements for the area of EZ defect across all time points was 0.90, indicating excellent reliability (**Fig 4B**). The average bias between measurements was 0.01 mm$^2$ (95% CI -0.18–0.20) (**Fig 4C**). Inter-rater reliability (ICC = 0.81) and agreement (average bias 0.07, 95% CI -0.17–0.31) for manual measurements between graders was good.

**Table 1. Comparison of ellipsoid zone defect area measurements.**

| Time Points | Manual | | | Automated | | | p-value | ICC | Average Bias (95% CI LOA) |
|---|---|---|---|---|---|---|---|---|---|
| | Minimum | Maximum | Mean ± SD | Minimum | Maximum | Mean ± SD | | | |
| All (mm$^2$) | 0.005 | 0.78 | 0.19 ± 0.23 | 0.005 | 0.76 | 0.19 ± 0.21 | 0.76 | 0.90 | 0.01 (-0.18–0.20) |
| M01 (mm$^2$) | 0.005 | 0.78 | 0.24 ± 0.24 | 0.005 | 0.76 | 0.23 ± 0.25 | 0.89 | 0.89 | 0.00 (-0.24–0.24) |
| M06 (mm$^2$) | 0.006 | 0.78 | 0.19 ± 0.22 | 0.007 | 0.76 | 0.21 ± 0.21 | 0.27 | 0.94 | 0.01 (-0.14–0.16) |
| M12 (mm$^2$) | 0.005 | 0.78 | 0.12 ± 0.18 | 0.005 | 0.66 | 0.10 ± 0.15 | 0.16 | 0.88 | 0.03 (-0.12–0.18) |

Abbreviations: M01 Month 1, M06 Month 6, M12 Month 12; ICC, intra-class correlation coefficient, CI confidence interval, LOA limits of agreement

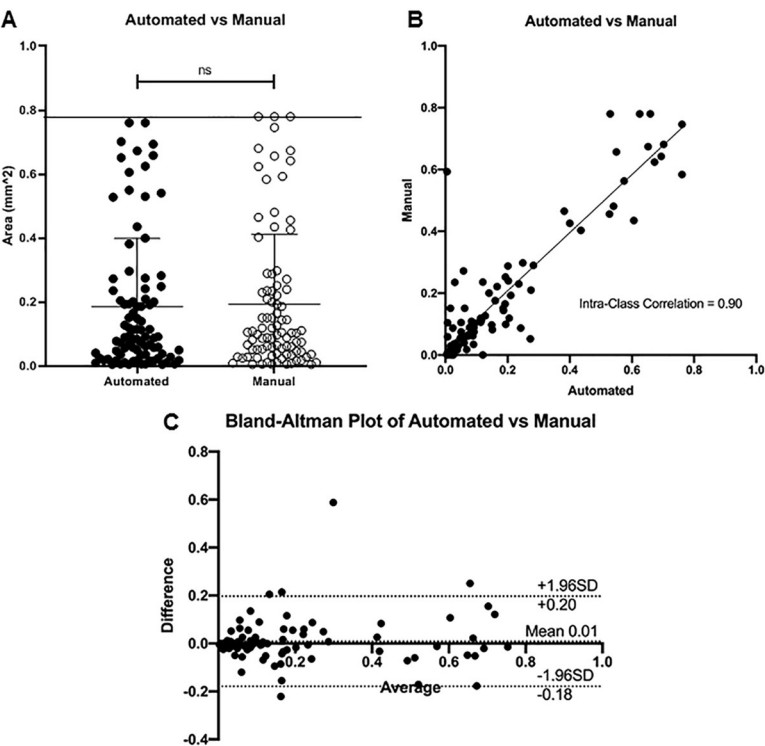

**Fig 4. Analysis of reliability and agreement between manual and automated area measurements of ellipsoid zone (EZ) defect. A.** Mean±SD area of EZ defect was similar between manual (0.19±0.23 mm$^2$) and automated (0.19±0.21 mm$^2$) measurements (p = 0.93). **B.** Intra-Class Correlation was 0.90 comparing area measurements of EZ defect. **C.** Bland-Altman Plot displaying differences against the averages between manual and automated area measurements of EZ defect. The average bias was 0.01 with a 95% confidence interval of the limits of agreement from -0.18–0.20. Abbreviation: ns not significant, SD standard deviation.

Area measurements did not differ between semi-automated and manual approaches at month 1 (p = 0.89), month 6 (p = 0.27), and month 12 (p = 0.16) (**Table 1**). The ICC at month 1 (0.89), month 6 (0.94), and month 12 (0.88) were good to excellent. The average bias between measurements was 0.00 (95% CI LOA: -0.24–0.24) at month 1, 0.01 (95% CI LOA: -0.14–0,16) at month 6, and 0.03 (95% CI LOA: -0.15–0.18) at month 12.

After segmentation of the EZ layer outside of the KNIME workflow, the time required to generate *en face* thickness maps from 184 XML files with EZ layer segmentation coordinates was 6 minutes 26 seconds with an average of 2.1 seconds per file. Using the semi-automated approach, the time required to identify and measure the EZ defect area within the CSF was 15 minutes 12 seconds with an average of 4.9 seconds per thickness map. Using the manual approach, the time required to identify and manually trace the EZ defect boundaries, and automatically quantify the area within the CSF was 2 hours 36 minutes with an average of 51 seconds per thickness map. The semi-automated approach reduced the time required to identify and quantify the EZ defect area within the CSF of 184 thickness maps by 2 hours with an average of 46.1 seconds per thickness map.

## Discussion

In this study, we developed a customized workflow that began with a semi-automated segmentation method for defining the EZ layer using nonproprietary software. These layer coordinates were used to then automatically generate *en face* thickness maps in eyes with macular

edema secondary to CRVO or HRVO in a randomly selected subgroup of SCORE2 subjects for both manual and semi-automatic segmentation of areas of EZ defect using the open-source software platform, KNIME. This semi-automated approach took advantage of machine-learning methods via the open-source Fiji[14] plugin Trainable Weka Segmentation (TWS)[13] and demonstrated excellent reliability and agreement when compared to manual measurements. These data suggest that our semi-automatic and open-source approach can be efficiently and reliably applied to similar workflows quantitatively assessing SD-OCT derived retinal morphology.

Quantitative assessment of the EZ commonly use manual measurements via the *en face* [26, 27] method, which consists of creating conforming segmentation lines along the upper and lower boundaries of the EZ band. The B-scan intensities through the segmented boundaries are projected to form an *en face* image with enhanced contrast between intact and absent EZ. [28] The EZ boundary is then traced to create a geographic representation. Manual measurement of the EZ is time intensive,[12] particularly in clinical trials with high volume data. Therefore, we sought to utilize accessible machine-learning tools to facilitate the identification of these regions.

The ability to measure the EZ is dependent on the visibility of the EZ layer. In the acute stages of RVO, evaluation of the EZ is impaired by the presence of hemorrhage and edema that reduces the outer retinal signal intensity.[27] At baseline, >90% of images were ungradable due to the presence of confounding hemorrhage and edema. Initial qualitative grading of the EZ was performed by expert graders at the SCORE2 central reading center (Fundus Photograph Reading Center, University of Wisconsin). The EZ was qualitatively deemed ungradable when the RPE was not visible due to signal blockage by hemorrhage and edema (**Fig 3A**). After intravitreal anti-VEGF treatment, 72% of available study eyes were gradable at month 1 for the evaluation of EZ integrity (**Fig 3B**). Similar approaches to evaluating EZ integrity exclude patients with severe edema, which has been defined as a CSF thickness greater than 600 μm.[5] Although we did not use this threshold, we did exclude EZ defect area measurements less than or equal to 0.004 mm$^2$. Area measurements less than this value were considered absent. We reviewed all data points outside the 95% CI of the LOA comparing the manual versus semi-automated measurements. Discrepancies between area measurements in all cases were due to poor image contrast as a result of impaired signal intensity from multiple factors, including media opacity, hemorrhage, and edema. These data suggest the semi-automated measurements did not disagree with those obtained by the manual approach, and semi-automated measures performed as well as manual outlines by expert graders (**Fig 5**).

This study has several limitations. We compared the manual to semi-automated area measurements of EZ defect using the mean area measurement and not by colocalizing the EZ defect areas generated from the two methods. This comparison does not capture the differences in position, orientation, and shape between the EZ boundary measurements. This measurement is impaired by the fact that similar EZ areas may have dissimilar EZ boundaries. Despite this limitation, the mean area measurement is similar to the difference in widths that have been used in reliability studies of EZ defect from individual B-scans in other disease processes, namely retinitis pigmentosa.[29] Additionally, because our study is limited to a smaller region of the CSF, we expect this difference to be minimal (**Fig 5**). The selection of scans for analysis of the EZ was performed by expert graders and was based on the SCORE2 SD-OCT grading protocol. However, all scans deemed of sufficient quality for EZ assessment (i.e. the RPE was visible) were evaluated by both the semi-automated and manual approaches. Future studies may examine the application of machine-learning for image selection. This analysis did not examine the correlation between the area of EZ defect with VA and other clinical data. Validation of our approach and association of EZ defect area measurements with clinical data

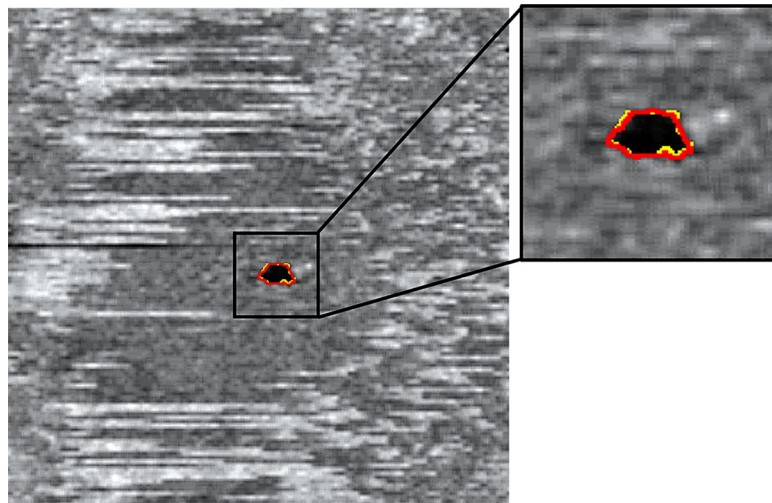

**Fig 5. Colocalization of ellipsoid zone (EZ) defect area identification between manual (yellow) and automated (red) approach.**

will be performed on the entire SCORE2 cohort upon study completion. Studies have examined the association of EZ integrity at baseline as a predictor of future visual acuity.[30–32] However, these studies included eyes with branch retinal vein occlusion, which presents with significantly less hemorrhage and fluid compared to central and hemi-retinal vein occlusion, especially within the CSF.[1] These studies also utilized high resolution SD-OCT scans, whereas the SCORE2 trial did not to improve the generalizability to the study methods and results. Although post-image processing to remove shadowing cast by hemorrhage and fluid has been performed,[27, 33] these studies have also utilized high resolution SD-OCT scans from a single SD-OCT device. Our method combined multiple SD-OCT machines. Finally, we examined EZ disruption in macular edema secondary to CRVO or HRVO. Future studies applying our methods to diabetic macular edema and other diseases may be beneficial.

There are many strengths of our approach. Our workflow combines multiple steps in the generation of EZ absence area measurements, such as EZ segmentation, export of files containing segmentation coordinates, generation of *en face* thickness maps that highlight the contrast between intact and absent EZ, and automatic delineation and quantification of EZ defects. The analysis utilized a classifier that acts as a computer-generated grader that can, once fully trained, measure regions of EZ defect, adding precision and high-throughput speed beyond that of manual tracing. Manual identification of EZ defect is time-consuming and error prone. In a study comparing methods of EZ boundary identification in retinitis pigmentosa, the average time required to delineate and trace the EZ boundary, not including segmentation of the EZ layer, was 4.1 minutes for the *en face* method.[12] Our semi-automated process is rapid, permitting the analysis of SD-OCT scan in seconds in a reliable and reproducible manner. Differences between manual EZ defect tracing times between the previously reported study and ours were likely due to area measurements only within the CSF.

## Conclusions

We developed a customized workflow using open-source software that applied manual and semi-automated methods for quantifying the area of EZ defect assessed by SD-OCT derived *en face* thickness maps in eyes with macular edema secondary to CRVO or HRVO. Semi-automated EZ defect area measurements were obtained through machine-learning and

demonstrated excellent agreement and reliability when compared to manual measurements, suggesting that our workflow may be applied to other quantitative assessments of SD-OCT derived retinal morphology. Our semi-automated approach will be validated in the entire SCORE2 cohort upon study completion.

## Acknowledgments

We would like to acknowledge the SCORE2 project manager Susan Reed and biostatistician Kyle W. McDaniel.

## Author Contributions

**Conceptualization:** Tyler Etheridge, Ellen T. A. Dobson, Chandana Papudesu, Ingrid U. Scott, Michael S. Ip, Kevin W. Eliceiri, Barbara A. Blodi, Amitha Domalpally.

**Data curation:** Tyler Etheridge, Ellen T. A. Dobson, Amitha Domalpally.

**Formal analysis:** Tyler Etheridge, Amitha Domalpally.

**Funding acquisition:** Ingrid U. Scott, Michael S. Ip, Barbara A. Blodi, Amitha Domalpally.

**Investigation:** Tyler Etheridge, Ellen T. A. Dobson, Marcel Wiedenmann, Chandana Papudesu, Ingrid U. Scott, Kevin W. Eliceiri, Barbara A. Blodi, Amitha Domalpally.

**Methodology:** Tyler Etheridge, Ellen T. A. Dobson, Marcel Wiedenmann, Chandana Papudesu, Kevin W. Eliceiri, Barbara A. Blodi, Amitha Domalpally.

**Project administration:** Tyler Etheridge, Ellen T. A. Dobson, Marcel Wiedenmann, Chandana Papudesu, Amitha Domalpally.

**Resources:** Ellen T. A. Dobson, Marcel Wiedenmann, Ingrid U. Scott, Michael S. Ip, Kevin W. Eliceiri, Barbara A. Blodi, Amitha Domalpally.

**Software:** Ellen T. A. Dobson, Marcel Wiedenmann, Kevin W. Eliceiri, Amitha Domalpally.

**Supervision:** Ingrid U. Scott, Michael S. Ip, Kevin W. Eliceiri, Barbara A. Blodi, Amitha Domalpally.

**Validation:** Tyler Etheridge, Chandana Papudesu, Amitha Domalpally.

**Visualization:** Tyler Etheridge, Marcel Wiedenmann, Amitha Domalpally.

**Writing – original draft:** Tyler Etheridge, Ellen T. A. Dobson, Amitha Domalpally.

**Writing – review & editing:** Tyler Etheridge, Ellen T. A. Dobson, Chandana Papudesu, Ingrid U. Scott, Michael S. Ip, Kevin W. Eliceiri, Barbara A. Blodi, Amitha Domalpally.

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
