## [Decision Letter · Decision Letter 0]

20 Jan 2020

PONE-D-19-34185

A Semi-Automated Machine-Learning Based Workflow for Ellipsoid Zone Analysis in Eyes with Macular Edema: SCORE 2 Pilot Study

PLOS ONE

Dear Dr Domalpally,

Thank you for submitting your manuscript to PLOS ONE. After careful consideration, we feel that it has merit but does not fully meet PLOS ONE’s publication criteria as it currently stands. Therefore, we invite you to submit a revised version of the manuscript that addresses the points raised during the review process.

The manuscript as reviewed by two exert reviewers and was found to be of interest, but they suggest some minor revisions. I agree and we look forward to the revised version 

We would appreciate receiving your revised manuscript by Mar 05 2020 11:59PM. To enhance the reproducibility of your results, we recommend that if applicable you deposit your laboratory protocols in protocols.io, where a protocol can be assigned its own identifier (DOI) such that it can be cited independently in the future. For instructions see: http://journals.plos.org/plosone/s/submission-guidelines#loc-laboratory-protocols

We look forward to receiving your revised manuscript.

Kind regards,

Demetrios G. Vavvas

Academic Editor

PLOS ONE

Journal Requirements:

2. Please amend your Data availability statement to describe how other researchers may obtain the datasets from the SCORE 2 study used in this study.

3. Please amend your Ethics statement to provide the approval numbers.

4. Please  amend your Competing interests statement to declare author commercial affiliations.

5. Thank you for including your ethics statement:  "The study was approved by the institutional review boards associated with each center and adhered to the tenets of the Declaration of Helsinki. All participants provided written informed consent."

7. Thank you for stating the following in the Financial Disclosure section:

"The Standard Care vs Corticosteroid for Retinal Vein Occlusion (SCORE) Study was supported by National Eye Institute (National Institutes of Health, Department of Health and Human Services) grants 5U10EY014351, 5U10EY014352, and 5U10EY014404; and Allergan, Inc."   

We note that you received funding from a commercial source: Allergan, Inc

8. Thank you for stating the following in the Financial Disclosure section:

"The Standard Care vs Corticosteroid for Retinal Vein Occlusion (SCORE) Study was supported by National Eye Institute (National Institutes of Health, Department of Health and Human Services) grants 5U10EY014351, 5U10EY014352, and 5U10EY014404; and Allergan, Inc."

We note that one or more of the authors have an affiliation to the commercial funders of this research study : KNIME GmbH.

Reviewers' comments:

Reviewer's Responses to Questions

**Comments to the Author**

1. Is the manuscript technically sound, and do the data support the conclusions?

Reviewer #1: Yes

Reviewer #2: Yes

2. Has the statistical analysis been performed appropriately and rigorously? 

Reviewer #1: Yes

Reviewer #2: Yes

3. Have the authors made all data underlying the findings in their manuscript fully available?

Reviewer #1: Yes

Reviewer #2: Yes

4. Is the manuscript presented in an intelligible fashion and written in standard English?

Reviewer #1: Yes

Reviewer #2: Yes

5. Review Comments to the Author

Reviewer #1: 1. A very well structured manuscript, technically sound, with data that support the conclusions.

2. The statistical analysis described is thorough.

3. The lack of EZ assessment at baseline scans defeats the purpose of using EZ integrity as a prognosticator of final visual outcome:

As described in the introduction, one of the reasons why EZ integrity is often assessed is its potential use as a predictive factor of the final visual outcome. In the present study, by converting cross sectional images into en face, hemorrhage or edema-associated signal blockage prevented the assessment of EZ integrity at baseline. (>90% of baseline images were ‘ungradable’)

Hence, as described, EZ integrity of only SD-OCT images at months 1, 6, and 12 was assessed, analysed and quantified in this study. Consequently, if a semi-automated machine learning system is used for the assessment of EZ integrity in en face images, the only prognostic information one can derive comes from the EZ integrity after intravitreal anti-VEGF treatment (at 1st month)

Reviewer #2: The paper by Tyler Etheridge et al. entitled 'A Semi-Automated Machine-Learning Based Workflow for Ellipsoid Zone Analysis in Eyes with Macular Edema: SCORE 2 Pilot Study' is an interesting study on the evaluation of the combination of a customized workflow with KNIME and machine learning tool for Fiji plugin: Trainable Weka Segmentation. The authors developed the customized workflow with semi-automated optical coherence tomography measurements and examined its agreement and reliability. And then, they found their new approach was reliable and time-saving. The study is well constructed. The obtained results will be of interest to researchers in the field. Their handy method using KNIME and Fiji plugin could be applied to other retinal diseases at a low cost. And I think the study has great potential not only in academic research but also in daily clinical practice.

I have few concerns,

(1) Page 7, line 170-172; what kind of dataset did the authors use for the learning process? From entire SCORE2 data? Or other normal subject data? And how much data did the authors use for training? Did the authors check if the performance reached the plateau or not?

(2) Page 11, line 247-255; how long did it take for the training process?

(3) Page 9, line 210-218; how did the authors judge whether each image could go through the analysis or not. Manually by graders or semi-automated with a program? Did 'gradable' mean manually measured? Did only images that could be manually measured go through semi-automated measurements? If the judgment is done by a human, not by a semi-automated program, then I suggest discussing this essential process in terms of the semi-automated approach limitation.

(4) Page 10, line 231-233 (Figure 3A); the authors evaluated the reliability with Bland-Altman plot in addition to intraclass correlation coefficient. But, I guess intraclass correlation coefficient is enough to evaluate both the reliability and agreement (ref. Terry K. Koo, Mae Y. Li. A Guideline of Selecting and Reporting Intraclass Correlation Coefficients for Reliability Research. J Chiropr Med. 2016 Jun; 15(2): 155–163. Published online 2016 Mar 31. doi: 10.1016/j.jcm.2016.02.012 PMCID: PMC4913118. Correction in: J Chiropr Med. 2017 Dec; 16(4): 346. Pubmed Central PMCID: PMC5731844).

(5) For the researchers and clinicians who want to use the workflow and semi-automated methods in their settings, is it possible to show the link of the programs or training dataset? If it's possible, it must be helpful for our fields.

6. PLOS authors have the option to publish the peer review history of their article (what does this mean?). If published, this will include your full peer review and any attached files.

Reviewer #1: No

Reviewer #2: No

---

## [Author Response · Author response to Decision Letter 0]

26 Mar 2020

Review Comments to the Author

Reviewer #1: 

1. A very well structured manuscript, technically sound, with data that support the conclusions

2. The statistical analysis described is thorough.

3. The lack of EZ assessment at baseline scans defeats the purpose of using EZ integrity as a prognosticator of final visual outcome:

As described in the introduction, one of the reasons why EZ integrity is often assessed is its potential use as a predictive factor of the final visual outcome. In the present study, by converting cross sectional images into en face, hemorrhage or edema-associated signal blockage prevented the assessment of EZ integrity at baseline. (>90% of baseline images were ‘ungradable’)

Hence, as described, EZ integrity of only SD-OCT images at months 1, 6, and 12 was assessed, analysed and quantified in this study. Consequently, if a semi-automated machine learning system is used for the assessment of EZ integrity in en face images, the only prognostic information one can derive comes from the EZ integrity after intravitreal anti-VEGF treatment (at 1st month)

Reply: No changes were made. Please see the discussion section (page 13, line 282-298). The inability to accurately assess the EZ on en face thickness maps due to reduced outer retinal signal intensity from blockage by hemorrhage and fluid is an inherent challenge in retinal vein occlusion. Without histopathological correlates it is difficult to know whether EZ defects on SD-OCT are artifacts resulting from signal blockage or loss of photoreceptors secondary to disease. However, our approach of assessing month 1 EZ integrity with subsequent visual acuity matches other studies referenced in the manuscript: 

Chan EW, Eldeeb M, Sun V, et al. Disorganization of Retinal Inner Layers and Ellipsoid Zone Disruption Predict Visual Outcomes in Central Retinal Vein Occlusion. Ophthalmol Retina 2019;3:83-92.

Tang F, Qin X, Lu J, Song P, Li M, Ma X. Optical Coherence Tomography Predictors of Short-Term Visual Acuity in Eyes with Macular Edema Secondary to Retinal Vein Occlusion Treated with Intravitreal Conbercept. Retina 2019.

Fujihara-Mino A, Mitamura Y, Inomoto N, Sano H, Akaiwa K, Semba K. Optical coherence tomography parameters predictive of visual outcome after anti-VEGF therapy for retinal vein occlusion. Clin Ophthalmol 2016;10:1305-13.

Reviewer #2: 

The paper by Tyler Etheridge et al. entitled 'A Semi-Automated Machine-Learning Based Workflow for Ellipsoid Zone Analysis in Eyes with Macular Edema: SCORE 2 Pilot Study' is an interesting study on the evaluation of the combination of a customized workflow with KNIME and machine learning tool for Fiji plugin: Trainable Weka Segmentation. The authors developed the customized workflow with semi-automated optical coherence tomography measurements and examined its agreement and reliability. And then, they found their new approach was reliable and time-saving. The study is well constructed. The obtained results will be of interest to researchers in the field. Their handy method using KNIME and Fiji plugin could be applied to other retinal diseases at a low cost. And I think the study has great potential not only in academic research but also in daily clinical practice.

I have few concerns,

(1) Page 7, line 170-172; what kind of dataset did the authors use for the learning process? From entire SCORE2 data? Or other normal subject data? And how much data did the authors use for training? Did the authors check if the performance reached the plateau or not?

Reply: No changes were made. Please see the methods section (page 8, line 175-179). The authors selected en face thickness maps that represented the inherent variability of the total dataset analyzed (184 scans). We chose approximately 10% of the dataset (12 en face thickness maps) for classifier training. Areas of normal EZ and EZ defects were manually traced using the Trainable Weka Segmentation plugin of Fiji as described by Arganda-Carreras et al.*. The authors sought to maximize the classifiers identification of regions of interest and minimize non-region of interest (false-positive) identification. A reliable and accurate classifier was trained with the initial 12 thickness maps, more would have been used if necessary, as is customary in such training protocols. 

* Arganda-Carreras I, Kaynig V, Rueden C, Eliceiri KW, Schindelin J, Cardona A, et al. Trainable Weka Segmentation: a machine learning tool for microscopy pixel classification. Bioinformatics. 2017;33(15):2424-6. doi: 10.1093/bioinformatics/btx180. PubMed PMID: 28369169

(2) Page 11, line 247-255; how long did it take for the training process?

Reply: No changes were made. The authors did not time the classifier training. However, after generation of the en face thickness maps and selection of images representing the heterogeneity of the image dataset, the training of the classifier required minimal time compared to manual annotations (20-30 minutes to obtain the first, most-reliable classifier with appropriate settings). 

(3) Page 9, line 210-218; how did the authors judge whether each image could go through the analysis or not. Manually by graders or semi-automated with a program? Did 'gradable' mean manually measured? Did only images that could be manually measured go through semi-automated measurements? If the judgment is done by a human, not by a semi-automated program, then I suggest discussing this essential process in terms of the semi-automated approach limitation.

Reply: The following changes were made (page 14, line 308-312). “The selection of scans for analysis of the EZ was performed by expert graders and was based on the SCORE2 SD-OCT grading protocol. However, all scans deemed of sufficient quality for EZ assessment (i.e. the RPE was visible) were evaluated by both the semi-automated and manual approaches. Future studies may examine the application of machine-learning for image selection.” 

(4) Page 10, line 231-233 (Figure 3A); the authors evaluated the reliability with Bland-Altman plot in addition to intraclass correlation coefficient. But, I guess intraclass correlation coefficient is enough to evaluate both the reliability and agreement (ref. Terry K. Koo, Mae Y. Li. A Guideline of Selecting and Reporting Intraclass Correlation Coefficients for Reliability Research. J Chiropr Med. 2016 Jun; 15(2): 155–163. Published online 2016 Mar 31. doi: 10.1016/j.jcm.2016.02.012 PMCID: PMC4913118. Correction in: J Chiropr Med. 2017 Dec; 16(4): 346. Pubmed Central PMCID: PMC5731844).

Reply: No changes were made. The authors sought to perform a rigorous comparison of the two approaches and subtle differences exist between reliability and agreement, which is described by Kottner et al.*.

*Kottner J, Streiner DL. The difference between reliability and agreement. J Clin Epidemiol. 2011 Jun;64(6):701-2; author reply 702. doi: 10.1016/j.jclinepi.2010.12.001. Epub 2011 Mar 16. PMID: 21411278.

(5) For the researchers and clinicians who want to use the workflow and semi-automated methods in their settings, is it possible to show the link of the programs or training dataset? If it's possible, it must be helpful for our fields.

Reply: The following changes were made to the Data Availability Statement. “The KNIME workflow is available at https://hub.knime.com/tetheridge/spaces/Public/latest/Raw_Thickness_Maps. The KNIME workflow protocol and training dataset is available at https://doi.org/10.6084/m9.figshare.11774577.”

---

## [Decision Letter · Decision Letter 1]

13 Apr 2020

PONE-D-19-34185R1

A Semi-Automated Machine-Learning Based Workflow for Ellipsoid Zone Analysis in Eyes with Macular Edema: SCORE2 Pilot Study

PLOS ONE

Dear Dr Domalpally,

Thank you for submitting your manuscript to PLOS ONE. After careful consideration, we feel that it has merit but does not fully meet PLOS ONE’s publication criteria as it currently stands. Therefore, we invite you to submit a revised version of the manuscript that addresses the points raised during the review process.

The manuscript has improved and there are some minor points remaining to be addressed. We look forward to the revised version 

We would appreciate receiving your revised manuscript by May 28 2020 11:59PM. To enhance the reproducibility of your results, we recommend that if applicable you deposit your laboratory protocols in protocols.io, where a protocol can be assigned its own identifier (DOI) such that it can be cited independently in the future. For instructions see: http://journals.plos.org/plosone/s/submission-guidelines#loc-laboratory-protocols

We look forward to receiving your revised manuscript.

Kind regards,

Demetrios G. Vavvas

Academic Editor

PLOS ONE

Reviewers' comments:

Reviewer's Responses to Questions

**Comments to the Author**

1. If the authors have adequately addressed your comments raised in a previous round of review and you feel that this manuscript is now acceptable for publication, you may indicate that here to bypass the “Comments to the Author” section, enter your conflict of interest statement in the “Confidential to Editor” section, and submit your "Accept" recommendation.

Reviewer #1: (No Response)

Reviewer #2: (No Response)

2. Is the manuscript technically sound, and do the data support the conclusions?

Reviewer #1: Yes

Reviewer #2: Yes

3. Has the statistical analysis been performed appropriately and rigorously? 

Reviewer #1: Yes

Reviewer #2: Yes

4. Have the authors made all data underlying the findings in their manuscript fully available?

Reviewer #1: Yes

Reviewer #2: Yes

5. Is the manuscript presented in an intelligible fashion and written in standard English?

Reviewer #1: Yes

Reviewer #2: Yes

6. Review Comments to the Author

Reviewer #1: The authors opted to make no changes on comment #3. Regarding their answer:

1. In their article entitled Spectral-domain optical coherence tomography (SD-OCT) patterns and response to intravitreal bevacizumab therapy in macular edema associated with branch retinal vein occlusion, Kang et al found that the integrity of the EZ before treatment could not be evaluated in cross sectional OCT scans in 8 of 67 patients with BRVO-ME.

Herein the authors by converting cross sectional images into en face report >90% of baseline images were ‘ungradable’. This is vastly different from the ungradable percentages in cross sectional OCT. Thus it seems the inability to accurately assess the EZ is not an inherent challenge in RVO (as the authors response to the comment suggests) rather an inherent limitation of en face images, which should be clearly stated in the limitations.

2. A prognosticator available at 1 month is not as useful to clinicians and patients as baseline prognosticators for long term functional outcomes in cases those exist. In BRVO the EZ integrity at baseline is a well established prognosticator that the en face approach seems to fail to offer.

All 3 articles that the authors cite in their response refer to cross sectional OCT studies and 2/3 actually did evaluate EZ disruption at baseline with no mention on the potential effect of macular edema. Hence, it seems that the en face transformation of cross sectional OCT images limits the ability to assess the EZ integrity due to macular edema and/or haemorrhage.

3. Further, the authors reply to the comment suggesting that histopathological studies would be needed to determine whether EZ defects on SD-OCT are artifacts resulting from signal blockage or loss of photoreceptors secondary to disease. However the article by Kanakis et al they are citing in the relevant part of their discussion (lines 282-298) describe the development of ‘a method of en face representation of the ellipsoid zone, along with the removal of shadows, to evaluate the ellipsoid layer disruption’ using ‘a MATLAB (Mathworks, Inc, Natick, MA) implementation of the algorithms for shadows removal’ concluding that ‘It could be assumed that the defects at the level of photoreceptors could be the result of the coexisting macular edema. However, the concordance of the ischemic area as shown in FA with the area of ellipsoid disruption at the en face reconstruction of the OCT makes this hypothesis unlikely.’

Good quality images are crucial in Machine Learning. The authors herein employed Machine Learning and transformed the cross sectional images to en face ending up >90% of baseline images were ‘ungradable’in terms of the main parameter the authors seek to study. To make their study stronger, and given their machine learning expertise, I would suggest the authors should embrace a similar approach as the en face OCT article by Kanakis et al they are citing and supplement their methods by developing an algorithm for artifact removal so that the quality control of the images they are feeding the Machine Learning is improved.

Reviewer #2: Almost all my concerns are answered clearly.

Just to be sure, please let me confirm it.

(1) Methods section (page 8, line 175-179). "Total dataset" comes from SCORE2 data. So, it didn't have normal subject data. Is it correct?

7. PLOS authors have the option to publish the peer review history of their article (what does this mean?). If published, this will include your full peer review and any attached files.

Reviewer #1: No

Reviewer #2: No

---

## [Author Response · Author response to Decision Letter 1]

14 Apr 2020

Response to editor : Our protocol is located with the KNIME workflow uploaded to the KNIME Hub and is completely open access. The link has provided in the data sharing component of the submission: https://hub.knime.com/tetheridge/spaces/Public/latest/Raw_Thickness_Maps

Review Comments to the Author

Reviewer #1: The authors opted to make no changes on comment #3. Regarding their answer:

1. In their article entitled Spectral-domain optical coherence tomography (SD-OCT) patterns and response to intravitreal bevacizumab therapy in macular edema associated with branch retinal vein occlusion, Kang et al found that the integrity of the EZ before treatment could not be evaluated in cross sectional OCT scans in 8 of 67 patients with BRVO-ME.

Herein the authors by converting cross sectional images into en face report >90% of baseline images were ‘ungradable’. This is vastly different from the ungradable percentages in cross sectional OCT. Thus it seems the inability to accurately assess the EZ is not an inherent challenge in RVO (as the authors response to the comment suggests) rather an inherent limitation of en face images, which should be clearly stated in the limitations.

Reply: Thank you for the comment. Both central and hemi-retinal vein occlusion have significantly more hemorrhage and fluid than branch retinal vein occlusion, making a comparison between the disease processes problematic. The SCORE2 study data were obtained from eyes with macular edema secondary to central or hemi-retinal vein occlusion. To further emphasize the challenge inherent in evaluating the ellipsoid zone on SD-OCT scans in central and hemi-retinal vein occlusion, regardless of cross sectional or en face evaluation, the authors have include a new figure (Figure 3 lines 217, 286, 288, and 478-481) depicting a representative cross sectional B-scan at baseline with significant hemorrhage and fluid blocking the outer retinal signal intensity to the point of being unable to identify the retinal pigment epithelium. The figure also includes a follow-up month 1 cross sectional image after resolution of a majority of hemorrhage and fluid with intravitreal anti-VEGF therapy and gradable ellipsoid zone. 

Change made to manuscript: Figure 3. (A) Representative center point SD-OCT scan deemed ungradable at baseline for ellipsoid zone (EZ) assessment. EZ was considered ungradable when the retinal pigment epithelium was not visible due to signal blockage by hemorrhage and edema. (B) Month 1 follow-up SD-OCT scan after single intravitreal anti-VEGF injection. EZ considered gradable. 

2. A prognosticator available at 1 month is not as useful to clinicians and patients as baseline prognosticators for long term functional outcomes in cases those exist. In BRVO the EZ integrity at baseline is a well established prognosticator that the en face approach seems to fail to offer.

All 3 articles that the authors cite in their response refer to cross sectional OCT studies and 2/3 actually did evaluate EZ disruption at baseline with no mention on the potential effect of macular edema. Hence, it seems that the en face transformation of cross sectional OCT images limits the ability to assess the EZ integrity due to macular edema and/or haemorrhage.

Reply: Thank you for the comment. The studies by Chan et al, Tang et al, and Fujihara-Mino et al included eyes with central retinal vein occlusion and branch retinal vein occlusion. As stated above, comparing the two distinct disease processes is problematic owing to the significant difference in hemorrhage and fluid in central retinal vein occlusion compared to branch retinal vein occlusion. In addition, these studies used high resolution SD-OCT scans, whereas the SCORE2 clinical trial did not to add generalizability to the study methods and results. This has been included in the discussion of our studies limitations. 

Change made to manuscript ( line 324): Studies have examined the association of EZ integrity at baseline as a predictor of future visual acuity.However, these studies included eyes with branch retinal vein occlusion, which presents with significantly less hemorrhage and fluid compared to central and hemi-retinal vein occlusion, especially within the CSF.

3. Further, the authors reply to the comment suggesting that histopathological studies would be needed to determine whether EZ defects on SD-OCT are artifacts resulting from signal blockage or loss of photoreceptors secondary to disease. However the article by Kanakis et al they are citing in the relevant part of their discussion (lines 282-298) describe the development of ‘a method of en face representation of the ellipsoid zone, along with the removal of shadows, to evaluate the ellipsoid layer disruption’ using ‘a MATLAB (Mathworks, Inc, Natick, MA) implementation of the algorithms for shadows removal’ concluding that ‘It could be assumed that the defects at the level of photoreceptors could be the result of the coexisting macular edema. However, the concordance of the ischemic area as shown in FA with the area of ellipsoid disruption at the en face reconstruction of the OCT makes this hypothesis unlikely.’

Good quality images are crucial in Machine Learning. The authors herein employed Machine Learning and transformed the cross sectional images to en face ending up >90% of baseline images were ‘ungradable’in terms of the main parameter the authors seek to study. To make their study stronger, and given their machine learning expertise, I would suggest the authors should embrace a similar approach as the en face OCT article by Kanakis et al they are citing and supplement their methods by developing an algorithm for artifact removal so that the quality control of the images they are feeding the Machine Learning is improved.

Reply: Thank you for the comment. Similar to the studies by Chan et al, Tang et al, and Fuijihara-Mino et al, the aforementioned study by Kanakis et al was performed using high resolution SD-OCT scans not obtained for use by clinical trials or clinical practice. The SCORE2 SD-OCT scans were not high-resolution. In addition, the study by Kanakis et al only utilized Spectralis SD-OCT machine. The SCORE2 trial utilized Spectralis and Cirrus SD-OCT machines. Therefore, our methods permitted the use of either imaging device. This limitation is included in the discussion (lines 320-321).

Change made to manuscript: These studies also utilized high resolution SD-OCT scans, whereas the SCORE2 trial did not to improve the generalizability to the study methods and results. Although post-image processing to remove shadowing cast by hemorrhage and fluid has been performed,these studies have also utilized high resolution SD-OCT scans from a single SD-OCT device. Our method combined multiple SD-OCT machines.

Reviewer #2: Almost all my concerns are answered clearly.

Just to be sure, please let me confirm it.

(1) Methods section (page 8, line 175-179). "Total dataset" comes from SCORE2 data. So, it didn't have normal subject data. Is it correct?

Reply: Thank you for the clarification. The total dataset comes from SCORE2 data and therefore does not include data from “normal” or “healthy” subjects. This clarification was made in the methods (lines 177-179). 

Change made to manuscript: All scans were obtained from the SCORE2 dataset and were therefore from eyes affected by either CRVO or HRVO.

---

## [Editor Report · Decision Letter 2]

16 Apr 2020

A Semi-Automated Machine-Learning Based Workflow for Ellipsoid Zone Analysis in Eyes with Macular Edema: SCORE2 Pilot Study

PONE-D-19-34185R2

Dear Dr. Domalpally,

We are pleased to inform you that your manuscript has been judged scientifically suitable for publication and will be formally accepted for publication once it complies with all outstanding technical requirements.

With kind regards,

Demetrios G. Vavvas

Academic Editor

PLOS ONE
---

## [Editor Report · Acceptance letter]

21 Apr 2020

PONE-D-19-34185R2 

A Semi-Automated Machine-Learning Based Workflow for Ellipsoid Zone Analysis in Eyes with Macular Edema: SCORE2 Pilot Study 

Dear Dr. Domalpally:

I am pleased to inform you that your manuscript has been deemed suitable for publication in PLOS ONE. Congratulations! Your manuscript is now with our production department. 

With kind regards,

on behalf of

Dr. Demetrios G. Vavvas 

Academic Editor

PLOS ONE